# Permo–Triassic boundary carbon and mercury cycling linked to terrestrial ecosystem collapse

Jacopo Dal Corso [1,2,4 ✉], Benjamin J. W. Mills [1,4 ✉], Daoliang Chu[2], Robert J. Newton [1], Tamsin A. Mather [3], Wenchao Shu[2], Yuyang Wu [2], Jinnan Tong[2] & Paul B. Wignall[1]

Records suggest that the Permo–Triassic mass extinction (PTME) involved one of the most severe terrestrial ecosystem collapses of the Phanerozoic. However, it has proved difficult to constrain the extent of the primary productivity loss on land, hindering our understanding of the effects on global biogeochemistry. We build a new biogeochemical model that couples the global Hg and C cycles to evaluate the distinct terrestrial contribution to atmosphere–ocean biogeochemistry separated from coeval volcanic fluxes. We show that the large short-lived Hg spike, and nadirs in $\delta^{202}$Hg and $\delta^{13}$C values at the marine PTME are best explained by a sudden, massive pulse of terrestrial biomass oxidation, while volcanism remains an adequate explanation for the longer-term geochemical changes. Our modelling shows that a massive collapse of terrestrial ecosystems linked to volcanism-driven environmental change triggered significant biogeochemical changes, and cascaded organic matter, nutrients, Hg and other organically-bound species into the marine system.

[1] School of Earth and Environments, University of Leeds, Leeds LS2 9JT, UK. [2] State Key Laboratory of Biogeology and Environmental Geology, China University of Geosciences, Wuhan 430074, China. [3] Department of Earth Sciences, University of Oxford, South Parks Road, Oxford OX1 3AN, UK. [4]These authors contributed equally: Jacopo Dal Corso, Benjamin J. W. Mills. ✉email: J.DalCorso@leeds.ac.uk; b.mills@leeds.ac.uk

The Permo–Triassic mass extinction (PTME) is the largest known extinction in Earth's history, with the loss of ~90% of species in the sea and ~70% of species on land[1–4]. The PTME has been causally linked to the emplacement of the Siberian Traps Large Igneous Province (LIP) and associated volcanic gas emissions (especially $CO_2$, $SO_2$ and halogens), via widespread environmental changes such as warming and oceanic anoxia[5–8]. The PTME also saw a crisis in terrestrial ecosystems, with loss of plant diversity, increased wildfire activity and consequent enhanced soil erosion[9–14]. Recent work has shown that the disruption of vegetation started before and culminated at the marine extinction level (Fig. 1), implying that the environmental disaster impacted terrestrial ecosystems first[12,14]. The cause of the terrestrial mass extinction is still unclear, and several kill mechanisms have been hypothesised. For example, a shift from a humid warm climate to an unstable highly seasonal climate and an associated increase in wildfires affected the equatorial Permo–Triassic peatlands, drastically reducing the abundance and diversity of the flora[14]; abnormal pollen and spores found in different localities around the world during the PTME interval suggest widespread mutagenesis possibly linked to an increase in UV-B radiation due to ozone depletion[15,16]; a terrestrial S-isotope record from the Karoo basin in South Africa could indicate volcanically driven acid rain at the P–T transition[17] that might have also severely impacted the flora. Whilst the taxonomic losses in terrestrial ecosystems are becoming clearer[12,18], and local, enhanced input of terrestrial material into marine environments has been recorded[9,11], the biogeochemical impacts and feedbacks on the exogenic C cycle are not known. It is possible that these impacts were severe: the PTME represents the largest, and maybe the only, known mass extinction of insects[19], suggesting that

there may have been a substantial decrease in available food sources at the lowest levels of the food chain.

The PTME is marked by an approximately two- to four-fold increase in marine sedimentary Hg concentration with respect to background levels during a ~400 kyr interval also characterised by negative $\delta^{13}C$ values[20], which implies a relatively long-term injection of Hg and $^{13}C$-depleted $CO_2$ into the atmosphere–land–ocean system during this time. Superimposed on this trend is a prominent, short-lived Hg spike, which is usually expressed as Hg/TOC, given the affinity of Hg with organic matter, and which is coincident with the collapse of the terrestrial ecosystems[14], the onset of the marine mass extinction interval and a sharp minimum in $\delta^{13}C$ values (Fig. 1). While higher Hg/TOC values have been reported preceding the marine extinction in the deep-water settings of Japan[21], these are an artefact of normalisation to values of TOC which are below the analytical detection limit (<<0.1%), and do not track increases in Hg concentrations (see TOC and Hg data in the Supplementary Information of ref. [21]). The Hg record is interpreted as evidence of increased Hg input into the Earth's surface system from the Siberian Traps[13,21–24]. However, Hg is also stored in large quantities in terrestrial biomass and soils[25–27]. The mobilisation of these terrestrial reservoirs during the PTME could also result in the increased loading of Hg to aquatic environments, even without an elevated volcanic Hg flux to the atmosphere[23,27–29]. Any such changes in the soil and biomass carbon reservoir would also have direct implications for the release of C to the atmosphere and the sedimentary $\delta^{13}C$ record.

Hg isotopes can be used to better understand how Hg has been transported into the sedimentary environment. Mass-independent fractionation (MIF—denoted by $\Delta$) occurs due to aqueous photoreduction of $Hg^{2+}$ to $Hg^0_{(g)}$ that takes place in surface waters and

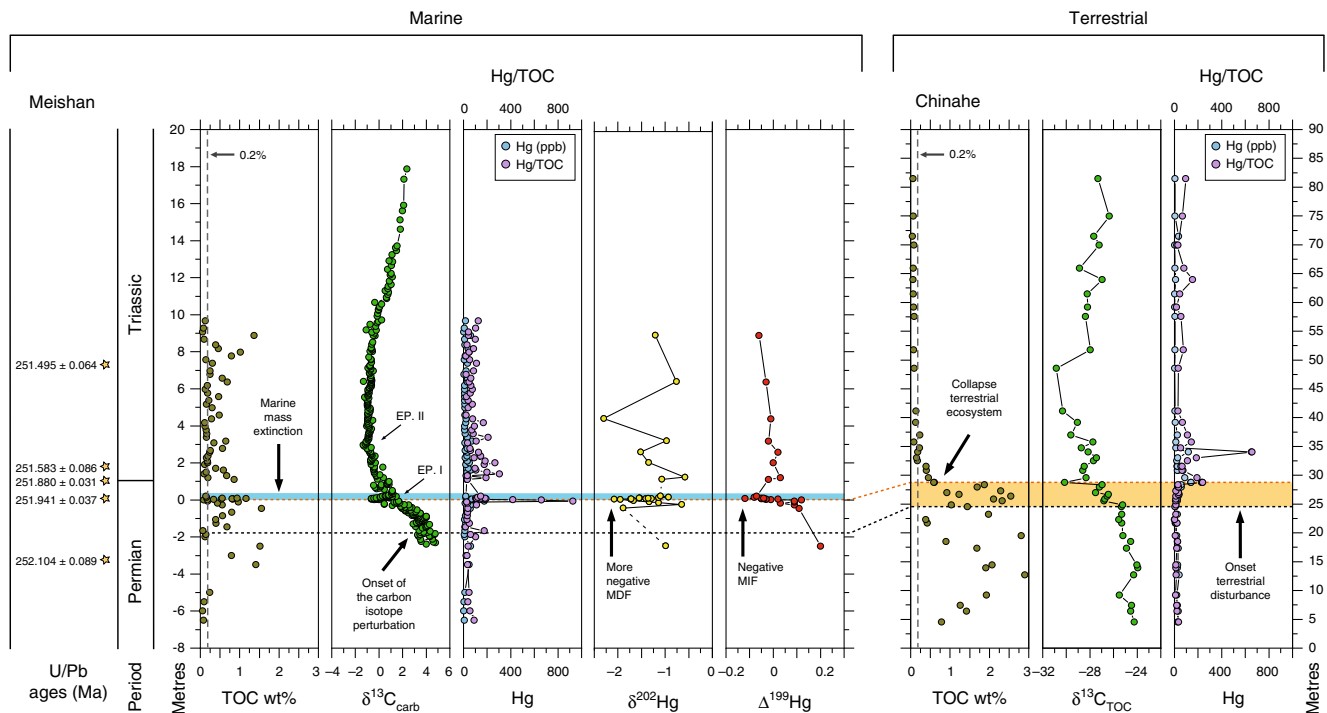

**Fig. 1 Permian–Triassic events.** Existing records show the terrestrial extinction started before, and its collapse culminated at, the marine extinction interval, where large Hg and Hg/TOC spikes, negative $\delta^{202}Hg$, negative $\Delta^{199}Hg$ and more negative $\delta^{13}C$ values (EP. I[30]) are recorded. After this event, the $\delta^{13}C$ record shows a second minimum (Ep. II), while Hg concentration decreases, but remains relatively higher than background values, and $\delta^{202}Hg$ and $\Delta^{199}Hg$ rebound towards more positive values. This pattern is seen in many P–T successions, in terrestrial and shallow- to deep-water settings. Meishan section: U-Pb ages from Burgess et al.[34]; TOC and C isotopes ($\delta^{13}C$) from Xie et al.[30]; Hg concentrations, $\delta^{202}Hg$ and $\Delta^{199}Hg$ data are from Grasby et al.[23] and Wang et al.[20]. Chinahe section: all data and correlation to Meishan section are from Chu et al.[14]. Note that to plot Hg (ppb) and Hg/TOC data, the same scale is used. MDF mass-dependent fractionation, MIF mass-independent fractionation, EP. episode.

in clouds[29]. This results in positive $\Delta^{199}$Hg values in the remaining water Hg$^{2+}$ pool, hence positive $\Delta^{199}$Hg values in sediments dominated by atmospheric Hg$^{2+}$ deposition. Conversely, slightly negative $\Delta^{199}$Hg values characterise the terrestrial reservoir (soil and biomass), which primarily captures Hg$^{0}_{(g)}$ [29]. During Hg uptake by plants additional mass-dependent fractionation (MDF) and MIF occur, resulting in more negative $\delta^{202}$Hg and $\Delta^{199}$Hg values[29]. The Permian–Triassic boundary shallow-water record of Meishan shows a prominent negative $\delta^{202}$Hg excursion in correspondence to the Hg spike coupled to a small negative $\Delta^{199}$Hg shift, but deeper water successions show persistent positive $\Delta^{199}$Hg[20,21,23,24]. Therefore, isotope data appear to indicate that the Hg was transported to the deep-water settings mostly via the atmosphere, and to the shallow-water settings via continental runoff and via the atmosphere[20,21,23,24].

The $\delta^{13}$C records at the Permian–Triassic boundary show two minima[30,31], which are here called EP. I and II (EP. = episode) following ref. [30] (Fig. 1). The observed $\delta^{13}$C trends are similarly recorded in different depositional settings and by different substrates (carbonates, bulk organic matter, separate plant remains)[14,30–32], strongly indicating that they represent actual changes in the C-isotope composition of the reservoirs of the exogenic C cycle. The Hg concentration spike occurs in the same interval as the minimum in $\delta^{13}$C values associated with the PTME (EP. I, Fig. 1). At Meishan, where the chronostratigraphic framework is well established[33,34], the initial large Hg spike occurs about 60 Kyrs after the onset of the carbon-isotope perturbation (Fig. 1). Data from non-marine end-Permian successions confirms this diachrony[13,14] (Fig. 1) and show that the Hg spike is also coincident with a sudden decrease in Total Organic Carbon (TOC) values to almost zero[14] (Fig. 1). The combination of geochemical and palaeontological data from these sections shows that the terrestrial ecosystem disruption started with the onset of the carbon-isotope perturbation and climaxed at the very sharp $\delta^{13}$C minimum (EP. I), coincident with the Hg spike (Fig. 1), and the start of the main marine extinction interval. This mismatch in both timings and fluxes between the C and Hg cycles at the PTME, suggest that the $^{13}$C-depleted C and the Hg came from multiple sources.

Overall, the interplay between volcanism and terrestrial reservoir changes in controlling PTME biogeochemistry is not well known, and previous attempts to model the $\delta^{13}$C record have been fundamentally hampered because of the lack of an independent tracer of the C source. To overcome this, we use published records of $\delta^{13}$C and Hg systematics to jointly constrain a new coupled C and Hg biogeochemical model. Our model shows that the large, sudden geochemical shifts at the PTME are best explained by a massive pulse of terrestrial biomass oxidation, while Siberian Traps volcanism can explain the longer-term geochemical changes.

## Results and discussion
**A coupled Hg-C cycle model.** Figure 2 shows the biogeochemical box model developed here. The full model derivation follows in the 'Methods' section. The model combines a multi-box sediment–ocean–atmosphere carbon-alkalinity cycle (based on previous work[35–37]), with the global mercury cycle[38,39]. The ocean is split into 'surface', 'high-latitude' and 'deep' boxes. It considers ocean circulation and carbonate speciation, and contains a simplified organic carbon cycle in which burial rates are prescribed. As well as computing the global C and Hg cycles it also computes $\delta^{13}$C of all C reservoirs and $\delta^{202}$Hg of the ocean reservoirs. A full atmosphere–ocean model of $\delta^{202}$Hg would require dynamic biosphere reservoirs, which would greatly increase model complexity.

We therefore simplify the system to a mixing model for marine $\delta^{202}$Hg, in which atmospheric and riverine inputs have different isotopic signatures. The atmosphere is assumed to have $\delta^{202}$Hg$_{atm}$ = −1 ‰, and riverine input is assumed to have $\delta^{202}$Hg$_{runoff}$ = −3 ‰, following ref. [40].

The model is set up for the late Permian by reducing the solar constant to that of 250 Ma, and increasing the background tectonic CO$_2$ degassing rate to 1.5 times the present day ($D$ = 1.5), in line with estimates for the Late Permian[41]. To obtain the observed pre-event ocean–atmosphere $\delta^{13}$C composition of ~3.5‰, we set the rate of land-derived organic carbon burial to 20% higher than present day, and adjust the composition of the weathered carbonate reservoir to 3‰. This is consistent with high terrestrial productivity and C burial in the Permian (e.g., coal forests and mires) and rapid recycling of more recently buried and $^{13}$C-enriched carbonate material.

In the following paragraphs, we test two model end-member scenarios: (I) the release of volcanic and thermogenic Hg and C from Siberian Traps activity alone, and (II) with the additional release of Hg and C as a consequence of the collapse of the terrestrial ecosystems.

**Volcanic and thermogenic degassing.** We first model the release of volcanic/volcanogenic Hg and C from the Siberian Traps. Existing radioisotope data show that the extinction, the negative $\delta^{13}$C excursion and Hg spike might have all occurred during the intrusive phase of the Siberian Traps[20,34,42]. It is suggested that the emplacement of large sills caused the combustion or thermal decomposition of organic-rich sediments with the consequent release of thermogenic volatiles, such as C and Hg[20,22]. It has been proposed that over a ~400 Kyrs intrusive phase the Siberian Traps emitted ~7600–13,000 Mg yr$^{-1}$ of volcanic Hg, which included both magmatic and coal-derived Hg[20,22,23]. Relating this Hg release to the background volcanic source is not straightforward because estimates of the background source vary, but taking the most likely present day range[25] (~90–360 Mg yr$^{-1}$), and further constraining this by taking into account the need to balance overall sedimentary burial of Hg (~190 Mg yr$^{-1}$), and the overall ~50% increase in tectonic degassing in the late Palaeozoic relative to today[43], we arrive at a best guess for the background late Permian Hg flux of ~300 Mg yr$^{-1}$. This means that the Siberian Traps eruption increased the geogenic Hg input by a factor of ~25–43 over ~400 Kyrs.

To test this scenario, we model Siberian Traps intrusion by increasing the volcanic Hg source by 25–43-fold for 450 Kyrs, while also increasing the CO$_2$ source in line with estimates[44,45] for Siberian Traps degassing based on magma volumes and sediment intrusion (by 4-8 × 10$^{12}$ moles/year). The CO$_2$ released by contact metamorphism at the PTME is assumed to have an average $\delta^{13}$C composition of −25‰[44]. Specifically, the input functions are:

$$f_{CO_{2input}} = \begin{bmatrix} -253 & -251.99 & -251.98 & -251.56 & -251.55 & -251 \end{bmatrix},$$
$$\begin{bmatrix} 0 & 0 & CO_{2ramp} & CO_{2ramp} & 0 & 0 \end{bmatrix}$$

$$F_{Hg_{input}} = \begin{bmatrix} -253 & -251.99 & -251.98 & -251.56 & -251.55 & -251 \end{bmatrix},$$
$$\begin{bmatrix} 1 & 1 & Hg_{ramp} & Hg_{ramp} & 1 & 1 \end{bmatrix}$$

Here the first vector is time in millions of years and the second is the flux alteration at that time. Here $CO_{2ramp}$ is the additional CO$_2$ release in mol yr$^{-1}$, and $Hg_{ramp}$ is the relative Hg degassing rate increase. For the duration of these pulses, the thermohaline circulation is also assumed to collapse due to warming and freshwater input[46]. We reduce the circulation term to 1 Sv over

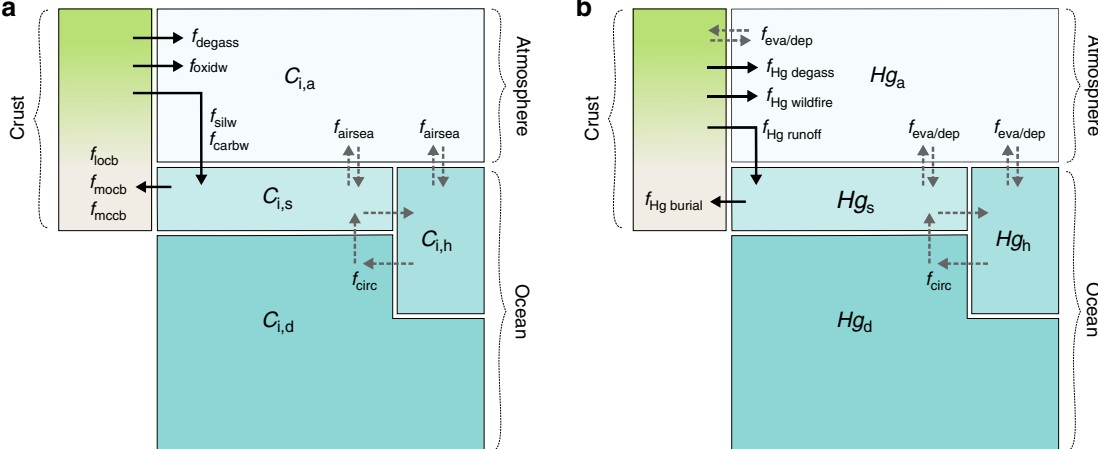

**Fig. 2 Model schematic.** Concentrations of modelled species are tracked in boxes representing the atmosphere (a), surface ocean (s), high-latitude ocean (h) and deep ocean (d). Exchange between boxes via air–sea exchange, circulation and mixing, or evasion and deposition are shown as dashed arrows. Biogeochemical fluxes between the hydrosphere and continents/sediments are shown as solid arrows. See text, 'Methods' and Supplementary Information for full details of fluxes. **a** C cycle. **b** Mercury cycle.

this period, which allows more rapid change in the model surface ocean C isotopes and Hg loading. This is a large reduction, and also reflects the simple structure of the model in which the entire low-latitude surface ocean is represented by a single box, and so is well-mixed. Figure 3a–d shows that this magnitude and timing of release of C and Hg is capable of driving the longer-term decline in carbonate $\delta^{13}$C, and the coeval long-term approximately two- to four-fold enrichment in shallow sediment Hg/TOC that is observed in Meishan. However, the model scenario does not capture the spike in Hg concentration, or nadir in $\delta^{13}$C (EP. I[30] in Fig. 1) that are coincident with the final stage of the terrestrial extinction. It also does not produce any substantial change in marine $\delta^{202}$Hg isotopes (Fig. 1d), because the primary Hg source to the ocean is the atmosphere for the full model run.

Within the model, we have also explored a scenario wherein the large Hg pulse represents a further rapid pulse of LIP volcanism. We have attempted this scenario in Supplementary Note 1 (Scenario I–2), where a 1 Kyr volcanic pulse is assumed to raise the Hg and C input rates by a further factor of 5. While the Hg/TOC can indeed be explained by an additional short-lived pulse of Hg, we require the total release rate of Hg to be ~200 times greater than background levels, and even then, this scenario fails to reproduce any of the Hg isotope signature or the nadir in carbonate $\delta^{13}$C (Supplementary Fig. 1).

**Terrestrial ecosystem collapse**. For scenario II, we explore the additional effects of a geologically rapid (~1 Kyr) pulse of Hg and C as the result of the collapse of terrestrial ecosystems at the PTME. The magnitude of this Hg flux is again difficult to quantify precisely, and we explore an increase of 100-fold over background conditions. This level of increase represents the magnitude required to drive the sedimentary Hg signal that we observe, and is compatible with the available terrestrial biosphere Hg reservoir: total soil Hg is estimated to be on the order of ~$10^6$ Mg Hg when considering a soil depth of ~15 cm[47]. So, our model Hg delivery flux would require decimetre-scale soil organic matter oxidation over 1000 years, coincident with the PTME and the sharp EP. I negative $\delta^{13}$C shift[11]. The Hg pulse is delivered directly to the low-latitude surface ocean via runoff in the model, and is accompanied by a pulse of 'soil oxidation' C which we assume raises the global rate of oxidative weathering by a factor of 30—a number chosen to

have the observed level of impact on the C-isotope record while being compatible with the Hg input change. We also assume a cessation of terrestrial organic C burial. Terrestrial Hg deposition and erosion is not altered during the pulse as the fluxes are minor by comparison. The new model functions applied in addition to the longer-term inputs of scenario I are:

$$F_{Cburial} = \begin{bmatrix} -253 & -251.951 & -251.950 & -251.949 & -251.948 & -251 \end{bmatrix}, \\ \begin{bmatrix} 1 & 1 & C_{ramp} & C_{ramp} & 1 & 1 \end{bmatrix}$$

$$F_{oxidw} = \begin{bmatrix} -253 & -251.951 & -251.950 & -251.949 & -251.948 & -251 \end{bmatrix}, \\ \begin{bmatrix} 1 & 1 & O_{ramp} & O_{ramp} & 1 & 1 \end{bmatrix}$$

$$F_{runoff} = \begin{bmatrix} -253 & -251.951 & -251.950 & -251.949 & -251.948 & -251 \end{bmatrix}, \\ \begin{bmatrix} 1 & 1 & Hg_{bio} & Hg_{bio} & 1 & 1 \end{bmatrix}$$

Here, the first vector is time in millions of years, and the second shows flux multipliers at these times. $C_{ramp}$, $O_{ramp}$ and $Hg_{bio}$ denote the relative rate of land organic C burial, oxidative weathering and Hg runoff, respectively, and are set at 0, 30 and 100, respectively, for the duration of the 1-kyr pulse.

This 'biosphere' pulse causes a short-term large concentration spike in the shallow marine Hg reservoir and its sediments, which is superimposed on the volcanically driven changes (Fig. 3e–h). The Hg spike is far larger than would be expected from simply increasing the volcanic source by the same amount because the biospheric Hg is delivered directly to the surface ocean and sedimentation occurs mostly on the shelf. With the inclusion of terrestrial C oxidation and cessation of terrestrial carbon burial, the model also replicates the transient shift to more negative $\delta^{13}$C values recorded at the marine extinction interval (EP. I[30,31] in Fig. 1): Terrestrial C oxidation is a source of isotopically light C[48]. The model now also shows a sharp negative $\delta^{202}$Hg shift in the shallow ocean box, which is triggered by increased Hg riverine input, but shows no change in the deeper ocean, where the source of Hg remains predominantly atmospheric. This also compares well with existing records (Fig. 3). At Meishan, which was located in the margins of the Yangtze carbonate platform, the Hg and Hg/TOC spike is coincident with more negative $\delta^{202}$Hg values (Fig. 1), while in the deeper water sections of south China the values are more positive[20,23].

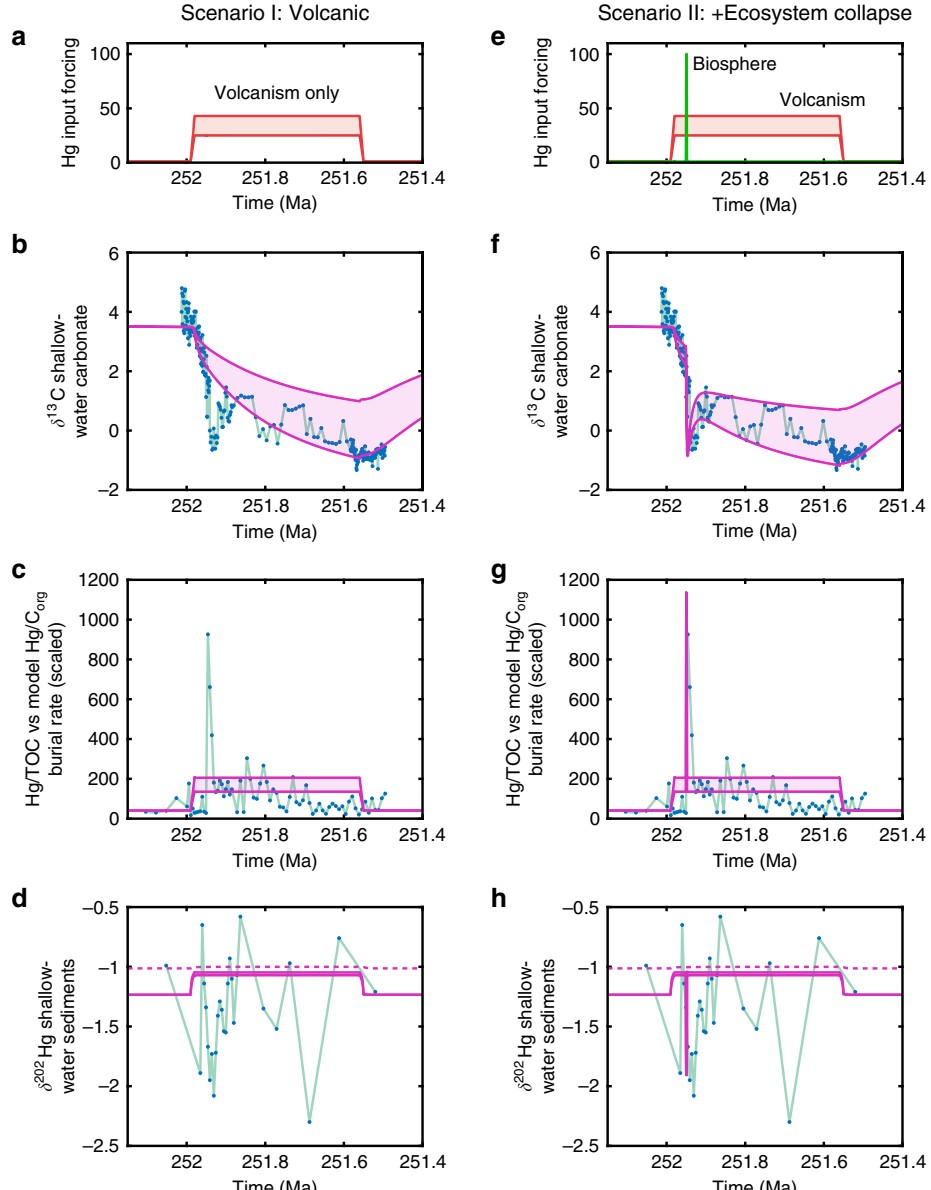

**Fig. 3 Model results. a–d** Volcanic C and Hg source only (thermogenic plus magmatic). **e–h** With addition of 'vegetation loss' Hg source and corresponding increase in organic carbon oxidation. C-isotope, Hg/TOC and δ202Hg data from 'Meishan GSSP' section[20,23]. In both columns, top panel (**a, e**) shows input forcing in terms of relative rate of Hg input, next panel (**b, f**) shows δ13C of new shallow ocean carbonate, third panel (**c, g**) shows Hg/TOC data versus the model molar Hg/Corg burial rate, which has been scaled to match the background as the model does not calculate weights. Final panel (**d, h**) shows model δ202Hg against data, where solid line is shallow water (box s) and dashed line is deep water (box d) Blue dots = published data; Red and green lines = model Hg fluxes; Pink lines = model output.

Hence, oxidation of terrestrial biomass is a compelling scenario to explain the palaeontological, sedimentological and geochemical data. There is clear observational evidence for the collapse of the terrestrial ecosystems and cessation of terrestrial C burial, stratigraphic evidence supporting the sequence and timing of the events (onset of the δ13C shift—collapse of the terrestrial ecosystem—Hg and C spike), sufficient quantity of Hg available, consistency with the isotopic evidence for changing Hg sources, and consistency with the δ13C records.

**Massive terrestrial biomass oxidation during the PTME.** Using our coupled C–Hg biogeochemical model, we show that the massive collapse of terrestrial ecosystems and oxidation of

terrestrial biomass during the Permian–Triassic extinction had a huge impact on global Hg and C biogeochemistry. Hg stored in the terrestrial reservoirs was rapidly released as a consequence of the loss of terrestrial biomass and increased soil erosion[9,14]. This mechanism is the best explanation for the sharp increased loading of Hg into both terrestrial and marine water bodies and the negative shift in δ202Hg in coincidence with the marine mass extinction. Contemporaneously, increased soil carbon oxidation introduced large quantities of isotopically light C, accounting for the sharp negative δ13C anomaly registered in the sedimentary record (EP. I[30]). In the model, the emission of Hg and C from magma and heating of sedimentary organic matter during the intrusive phase of the Siberian Traps LIP emplacement can account for the smaller,

two- to fourfold increase of Hg concentrations with respect to background levels, and the relatively longer negative $\delta^{13}C$ trend that is recorded by both carbonates and organic matter, in marine and terrestrial settings.

A new scenario emerges for the PTME that links the collapse of ecosystems on land to the global geochemical changes recorded at the marine extinction interval. The disruption of terrestrial environments started during the initial phases of the Siberian Traps emplacement likely due to the release of volcanic gases as $CO_2$, $SO_2$ and halogens, which could have triggered acid rain, ozone depletion, volcanic darkness, rapid cooling and subsequent global warming[8,49]. At the culmination of the terrestrial disturbance interval, when the ecosystems totally collapsed, large amounts of $^{13}C$-depleted C and Hg deriving from a massive oxidation of terrestrial biomass were transported into aqueous habitats causing a steep decline in sedimentary $\delta^{13}C$ (carbonates and organic matter), a sedimentary Hg concentrations spike and a shift in $\delta^{202}Hg$ (Fig. 3). At this level, the marine mass extinction started. This, according to the existing chronostratigraphic framework, happened ~60 Kyrs after the onset of the carbon-isotope perturbation and of the terrestrial ecological disturbances[14] (Fig. 1).

The biogeochemical cycle of Hg is intimately linked to the cycle of organic matter and its constituting elements, such as C, N, S and P[50]. Hence, besides Hg and C, other organically-bound species would have been transferred from the terrestrial reservoirs into the marine system in large quantities at EP. I (Fig. 1). Addition of these species, particularly the nutrients P and N, are easily capable of driving ecosystem turnover, anoxia and eutrophication, and it is likely that this terrestrial input contributed to the marine extinction[9,11]. Our model does not include these additional cycles, but other models have shown that a relatively small increase in marine P delivery (2–3-fold) has the potential to drive marine anoxia or euxinia[51,52]. The scale of the terrestrial ecosystem collapse at the PTME could explain the severity of the biotic crisis at the Permian–Triassic boundary at all trophic levels, and should be a key consideration for future research. For other events, the Hg records are not so consistent nor as detailed as for the PTME. However, it is very likely that future research on other intervals could show the same Hg and C patterns as for the PTME.

## Methods

**Model derivation**. This model is designed to track the transfer and isotopic signature of atmospheric and marine carbon and mercury over geological time, while being broadly applicable to changes on the timescale of ocean circulation. The biogeochemical system is taken largely from ref. [36], with some additions from refs. [37,53,54], with the underlying hydrological model from ref. [35]. The Hg cycle follows ref. [25].

**Model structure**. The model has three ocean boxes: surface (s), high latitude (h) and deep (d). As in ref. [35], the surface box is 100-m deep and occupies 85% of the ocean surface, whereas the high-latitude box is 250-m deep and represents 15% of the ocean surface. Each ocean box includes the same biogeochemical species, and a thermohaline circulation mixes the boxes in the order s, h, d. The upper boxes exchange with the atmosphere, which is a single box. As well as transfer fluxes between ocean and atmosphere boxes, biogeochemical fluxes of weathering, degassing and burial operate between the surface system and crust.

**Model species**. All model species are shown in Table 1.

**Model fluxes**. Model fluxes, with equations and present values are shown in Table 2.

**Non-flux calculations**. Atmospheric $CO_2$ volume ratio is calculated as:

$$CO_2ppm = 280\frac{CO_{2a}}{CO_{2a_0}}$$

where $CO_{2a}$ is atmospheric $CO_2$ in moles, and $CO_{2a_0}$ is this value at present day.

Global average surface temperature (GAST) is:

$$GAST = 288 + k_{clim}\left(\frac{\log\left(\frac{CO_2ppm}{280}\right)}{\log(2)}\right) - 7.4\left(\frac{t_{geol}}{-570}\right)$$

where $k_{clim}$ is climate sensitivity to doubling $CO_2$, and $t_{geol}$ is time in millions of years before present and is expressed in negative terms. Low-latitude surface temperature ($T_s$) is assumed to scale by $\frac{2}{3}$ times global temperature change, and both high-latitude ($T_h$) and deep ($T_d$) temperature are assumed to follow global temperature change.

For carbonate speciation, effective equilibrium constants are calculated following refs. [36,55]:

$$K_{carb} = 5.75\times10^{-4} + 6\times10^{-6}(T_j - 278)$$

$$K_{CO_2} = 0.035 + 0.0019(T_j - 278)$$

Dissolved carbon species are then calculated following Walker and Kasting[36]:

$$[HCO_3^-]_j = \frac{DIC_j - \sqrt{DIC_j^2 - ALK_j\left(2DIC_j - ALK_j\right)\left(1 - 4K_{carb}\right)}}{1 - 4K_{carb}}$$

**Table 1 Model species.**

| Description | Name | Exists in | Size at present |
|---|---|---|---|
| Surface ocean water | $W_s$ | Surface ocean | $3.07\times10^{16}$ m³ |
| High-latitude water | $W_h$ | High latitude | $1.35\times10^{16}$ m³ |
| Deep water | $W_d$ | Deep ocean | $1.35\times10^{18}$ m³ |
| Atmospheric $CO_2$ | $CO_{2a}$ | Atmosphere | $5\times10^{16}$ mol C |
| Surface ocean DIC | $DIC_s$ | Surface ocean | $6\times10^{16}$ mol C* |
| High-latitude DIC | $DIC_h$ | High latitude | $3\times10^{16}$ mol C* |
| Deep ocean DIC | $DIC_d$ | Deep ocean | $3\times10^{18}$ mol C* |
| Surface ocean alkalinity | $ALK_s$ | Surface ocean | $6\times10^{16}$ mol CaCO₃ equiv.* |
| High-latitude alkalinity | $ALK_h$ | High latitude | $3\times10^{16}$ mol CaCO₃ equiv.* |
| Deep ocean alkalinity | $ALK_d$ | Deep ocean | $3\times10^{18}$ mol CaCO₃ equiv.* |
| $\delta^{13}C$ of atmospheric $CO_2$ | $\delta^{13}CO_{2a}$ | Atmosphere | $-7$ ‰ |
| $\delta^{13}C$ of surface ocean DIC | $\delta^{13}DIC_s$ | Surface ocean | 0.1 ‰** |
| $\delta^{13}C$ of high-latitude DIC | $\delta^{13}DIC_h$ | High latitude | 0.1 ‰** |
| $\delta^{13}C$ of deep ocean DIC | $\delta^{13}DIC_d$ | Deep ocean | 0.1 ‰** |
| Atmospheric Hg | $Hg_a$ | Atmosphere | $3.5\times10^{6}$ mol Hg*** |
| Surface ocean Hg | $Hg_s$ | Surface ocean | $1.34\times10^{7}$ mol Hg*** |
| High-latitude Hg | $Hg_h$ | High latitude | $5.9\times10^{6}$ mol Hg*** |
| Deep ocean Hg | $Hg_d$ | Deep ocean | $5.9\times10^{8}$ mol Hg*** |

*Starting values chosen close to equilibrium values, model equilibrates to DIC ≈ 2 mM and ALK ≈ 2.2 mM.
**Note that 0.1 is used instead of 0 to increase model stability at initialisation *** after Amos et al.[25] and scaled to ocean box volumes.

**Table 2 Model fluxes.**

| Description | Name | Equation | Size at present |
|---|---|---|---|
| Transfer fluxes | $tran_{ij}$ | $C_i f_{circ}$ | Multiple |
| Air–sea exchanges | $f_{airsea_j}$ | $A_j M_{atm} \left(\frac{pCO_{2a} - pCO_{2j}}{\tau_{oa}}\right)$ | Multiple |
| Silicate weathering | $f_{silw}$ | $k_{basw} f_{T_{bas}} + k_{granw} f_{T_{gran}}$ | $8 \times 10^{12}$. mol C yr$^{-1}$ |
| Carbonate weathering | $f_{carbw}$ | $k_{carbw} f_{Tcarb}$ | $8 \times 10^{12}$ mol C yr$^{-1}$ |
| Oxidative weathering | $f_{oxidw}$ | $k_{oxidw} (RO_2)^{0.5} F_{oxidw}$ | $7.75 \times 10^{12}$ mol C yr$^{-1}$ |
| Carbonate degassing | $f_{ccdeg}$ | $k_{ccdeg} D$ | $8 \times 10^{12}$ mol C yr$^{-1}$ |
| Organic carbon degassing | $f_{ocdeg}$ | $k_{ocdeg} D$ | $1.25 \times 10^{12}$ mol C yr$^{-1}$ |
| Mare carbonate burial | $f_{mccb}$ | $k_{mccb} \frac{(\Omega-1)^{1.7}}{\Omega_0}$ | $16 \times 10^{12}$ mol C yr$^{-1}$ |
| Marine organic C burial | $f_{mocb}$ | $k_{mocb}$ | $4.5 \times 10^{12}$ mol C yr$^{-1}$ |
| Marine carbonate burial | $f_{mccb}$ | $k_{mccb}$ | $16 \times 10^{12}$ mol C yr$^{-1}$ |
| Land organic C burial | $f_{locb}$ | $k_{locb} F_{Cburial}$ | $4.5 \times 10^{12}$ mol C yr$^{-1}$ |
| Volcanic Hg release | $f_{Hg_{volc}}$ | $k_{Hg_{volc}} F_{Hg_{input}}$ | $1.5 \times 10^{6}$ mol Hg yr$^{-1}$* |
| Wildfire Hg release | $f_{Hg_{wildfire}}$ | $k_{Hg_{wildfire}}$ | $5 \times 10^{5}$ mol Hg yr$^{-1}$* |
| Riverine Hg runoff | $f_{Hg_{runoff}}$ | $k_{Hg_{runoff}} F_{runoff}$ | $2 \times 10^{6}$ mol Hg yr$^{-1}$* |
| Marine Hg burial | $f_{Hg_b}$ | $k_{Hg_b} \left(\frac{Hg_s}{Hg_{s_0}}\right)$ | $2 \times 10^{6}$ mol Hg yr$^{-1}$* |
| Vegetation Hg deposition | $f_{vegdep}$ | $k_{Hg_{vegdep}} \left(\frac{Hg_a}{Hg_{a_0}}\right)$ | $1 \times 10^{7}$ mol Hg yr$^{-1}$* |
| Vegetation Hg evasion | $f_{vegeva}$ | $k_{Hg_{vegeva}}$ | $1 \times 10^{7}$ mol Hg yr$^{-1}$* |
| Ocean Hg deposition | $f_{oceandep_j}$ | $k_{Hg_{oceandep}} \left(\frac{Hg_a}{Hg_{a_0}}\right)$ | $1.5 \times 10^{7}$ mol Hg yr$^{-1}$* |
| Ocean Hg evasion | $f_{oceaneva_j}$ | $k_{Hg_{oceaneva}} \left(\frac{Hg_j}{Hg_{j_0}}\right)$ | $1.5 \times 10^{7}$ mol Hg yr$^{-1}$* |

*Volcanic Hg flux follows ref. [25], and is assumed equivalent to the total Hg burial in their model to close the system over long timescales. This value is within their stated reasonable range but is ~2× their chosen volcanic flux, the value is further increased by 50% due to higher rates of tectonic degassing through the Permian and Triassic. Runoff and burial fluxes are calculated to close the system under volcanic and wildfire input. Deposition and evasion fluxes directly follow ref. [25].

$$[CO_3^{2-}]_j = \frac{ALK_j - [HCO_3^-]_j}{2}$$

$$pCO_{2j} = \frac{K_{CO_2}[HCO_3^-]^2}{[CO_3^{2-}]}$$

We also explicitly calculate [H$^+$] concentration to observe model pH:

$$[H^+] = K_2 \frac{[HCO_3^-]}{[CO_3^{2-}]}$$

Calcium carbonate saturation state is calculated as:

$$\Omega_j = \frac{[Ca]_j [CO_3^{2-}]_j}{K_{sp}}$$

where $\Omega_j$ is the CaCO$_3$ saturation state in box $j$, and $K_{sp}$ is the solubility product.

For terrestrial chemical weathering, temperature dependence of basalt and granite weathering is calculated as:

$$f_{Tbas} = e^{0.0608(GAST - 288)} (1 + 0.038(GAST - 288))^{0.65}$$

$$f_{Tgran} = e^{0.0724(GAST - 288)} (1 + 0.038(GAST - 288))^{0.65}$$

And temperature dependence of carbonate weathering:

$$f_{Tcarb} = 1 + 0.087(GAST - 288)$$

**Fixed parameters**. Fixed parameters are shown in Table 3.

**Differential equations**. The following equations track the 11 non-water species from Table 1.
Atmospheric CO$_2$:

$$\frac{d(CO_{2a})}{dt} = -f_{airsea_s} - f_{airsea_h} + f_{ccdeg} + f_{ocdeg} + f_{oxidw} - f_{locb} - f_{carbw} - 2f_{silw} + f_{CO_{2input}}$$

Low-latitude surface ocean DIC:

$$\frac{d(DIC_s)}{dt} = f_{airsea_s} + tran_{DIC_{ds}} - tran_{DIC_{sh}} + 2f_{carbw} + 2f_{silw} - f_{mccb} - f_{mocb}$$

High-latitude surface ocean DIC:

$$\frac{d(DIC_h)}{dt} = f_{airsea_h} + tran_{DIC_{sh}} - tran_{DIC_{hd}}$$

**Table 3 Fixed model parameters.**

| Description | Name | Size at present |
|---|---|---|
| Thermohaline speed | $f_{circ}$ | 10 Sv |
| Relative area of low-latitude surface ocean | $A_s$ | 0.85 |
| Relative area of high-latitude surface ocean | $A_h$ | 0.15 |
| Present day moles of atmospheric CO$_2$ | $M_{atm}$ | $5 \times 10^{16}$ mol C |
| Timescale parameter for gas exchange | $\tau_{oa}$ | 10 years |
| Long-term climate sensitivity | $k_{clim}$ | 5 K |
| CO$_2$ second dissociation constant | $K_2$ | $7.4 \times 10^{-10}$ |
| Calcium carbonate solubility product | $K_{sp}$ | 0.8 mmol$^2$ kg$^{-2}$ * |
| Present day CaCO$_3$ saturation state | $\Omega_0$ | 3 |
| Marine calcium concentration | $[Ca]$ | 10 mM |
| Photosynthetic C fractionation | $\Delta B$ | 27‰ |
| C-isotope composition of carbonates | $\delta^{13}C_C$ | 0‰ |
| C isotope composition of organics | $\delta^{13}C_G$ | −27‰ |

*Chosen within ocean range (0.43–1.15)[56] to achieve reasonable DIC and ALK at present.

Deep ocean DIC:

$$\frac{d(DIC_d)}{dt} = tran_{DIC_{hd}} - tran_{DIC_{ds}}$$

Low-latitude surface ocean alkalinity:

$$\frac{d(ALK_s)}{dt} = tran_{ALK_{ds}} - tran_{ALK_{sh}} + 2f_{carbw} + 2f_{silw} - 2f_{mccb}$$

High-latitude surface ocean alkalinity:

$$\frac{d(ALK_h)}{dt} = tran_{ALK_{sh}} - tran_{ALK_{hd}}$$

Deep ocean alkalinity:

$$\frac{d(ALK_d)}{dt} = tran_{ALK_{hd}} - tran_{ALK_{ds}}$$

$\delta^{13}C$ of atmospheric $CO_2$:

$$\frac{d\left(\delta^{13}CO_{2a} \cdot CO_{2a}\right)}{dt} = -f_{airsea_s}\delta^{13}C_{atm} - f_{airsea_h}\delta^{13}C_{atm} + f_{ccdeg}\delta^{13}C_C$$
$$+ f_{ocdeg}\delta^{13}C_G + f_{oxidw}\delta^{13}C_G - f_{locb}\left(\delta^{13}C_{atm} - \Delta B\right)$$
$$- f_{carbw}\delta^{13}C_{atm} - 2f_{silw}\delta^{13}C_{atm} + f_{CO_{2input}}\delta^{13}C_{input}$$

$\delta^{13}C$ of low-latitude surface ocean DIC:

$$\frac{d\left(\delta^{13}DIC_s \cdot DIC_s\right)}{dt} = f_{airsea_s}\delta^{13}C_{atm} + tran_{DIC_{ds}}\delta^{13}DIC_d - tran_{DIC_{sh}}\delta^{13}DIC_s$$
$$+ f_{carbw}\delta^{13}C_{atm} + f_{carbw}\delta^{13}C_C + 2f_{silw}\delta^{13}C_{atm}$$
$$- f_{mccb}\delta^{13}DIC_s - f_{mocb}\left(\delta^{13}DIC_s - \Delta B\right)$$

$\delta^{13}C$ of high-latitude surface ocean DIC:

$$\frac{d\left(\delta^{13}DIC_h \cdot DIC_h\right)}{dt} = f_{airsea_h}\delta^{13}C_{atm} + tran_{DIC_{sh}}\delta^{13}DIC_s - tran_{DIC_{hd}}\delta^{13}DIC_h$$

$\delta^{13}C$ of deep ocean DIC:

$$\frac{d\left(\delta^{13}DIC_d \cdot DIC_d\right)}{dt} = tran_{DIC_{hd}}\delta^{13}DIC_h - tran_{DIC_{ds}}\delta^{13}DIC_d$$

Atmospheric Hg:

$$\frac{d\left(Hg_a\right)}{dt} = f_{Hg_{volc}} + f_{Hg_{wildfire}} - f_{oceandep_h} + f_{oceaneva_h}$$
$$- f_{oceandep_s} + f_{oceaneva_s} - f_{vegdep} + f_{vegeva}$$

Low-latitude surface ocean Hg:

$$\frac{d\left(Hg_s\right)}{dt} = f_{Hg_{runoff}} + f_{oceandep_s} - f_{oceaneva_s} + tran_{Hg_{ds}} - tran_{Hg_{sh}} - f_{Hg_b}$$

High-latitude surface ocean Hg:

$$\frac{d\left(Hg_h\right)}{dt} = f_{oceandep_h} - f_{oceaneva_h} + tran_{Hg_{sh}} - tran_{Hg_{hd}}$$

Deep ocean Hg:

$$\frac{d\left(Hg_d\right)}{dt} = tran_{Hg_{hd}} - tran_{Hg_{ds}}$$

$\delta^{202}Hg$ of Low-latitude surface ocean Hg:

$$\frac{d\left(\delta^{202}Hg_s \cdot Hg_s\right)}{dt} = f_{Hg_{runoff}}\delta^{202}Hg_{runoff} + f_{oceandep_s}\delta^{202}Hg_{atm} - f_{oceaneva_s}\delta^{202}Hg_s$$
$$+ tran_{Hg_{ds}}\delta^{202}Hg_d - tran_{Hg_{sh}}\delta^{202}Hg_s - f_{Hg_b}\delta^{202}Hg_s$$

$\delta^{202}Hg$ of High-latitude surface ocean Hg:

$$\frac{d\left(\delta^{202}Hg_h \cdot Hg_h\right)}{dt} = f_{oceandep_h}\delta^{202}Hg_{atm} - f_{oceaneva_h}\delta^{202}Hg_h$$
$$+ tran_{Hg_{sh}}\delta^{202}Hg_s - tran_{Hg_{hd}}\delta^{202}Hg_h$$

$\delta^{202}Hg$ of Deep ocean Hg:

$$\frac{d\left(\delta^{202}Hg_d \cdot Hg_d\right)}{dt} = tran_{Hg_{hd}}\delta^{202}Hg_h - tran_{Hg_{ds}}\delta^{202}Hg_d$$

## Data availability
The geochemical data used in this paper come from already published literature, as cited in the text.

## Code availability
MATLAB code to run the model is available from B.J.W. Mills on request.

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

## Acknowledgements

J.D.C. thanks Timothy M. Lenton for useful comments from which this study emerged. J.D.C., R.J.N. and P.W. acknowledge support from NERC grant NE/P013724/1. J.D.C. also acknowledges the One Hundred Talent Program of China University of Geosciences (CUG) Wuhan, China. B.J.W.M. acknowledges support from NERC grants NE/S009663/1 and NE/R010129/1 and from a University of Leeds Academic Fellowship. D.C., J.T., W.S. and Y.W. acknowledge National Natural Science Foundation of China grants (grants 41530104, 41661134047). T.A.M. acknowledges funding from ERC consolidator grant (ERC-2018-COG- 818717 -V-ECHO).

## Author contributions

J.D.C. conceived the study. B.J.W.M. built the box model. J.D.C. and B.J.W.M. designed the model scenarios with in-depth inputs from T.A.M., P.B.W., D.C. and R.J.N. J.D.C., D.C., W.S., Y.W. and P.B.W. compiled and discussed the geochemical and chronostratigraphic data. All authors discussed the results and contributed to the writing of the paper. P.B.W., R.J.N. and J.T. provided the funding.

## Competing interests

The authors declare no competing interests
