## [Peer Review File · Nature Communications]

Reviewers' comments:

Reviewer #1 (Remarks to the Author):

Review of Dal Corso et al. "Permo-Triassic boundary carbon and mercury cycling linked to terrestrial ecosystem collapse" for Nature Communications

In this modeling study, Dal Corso et al. test several hypotheses regarding causes and timing of the end-Permian mass extinction event by focusing on the environmental effects of massive volcanic loading and terrestrial biosphere oxidation, as reflected in the global carbon and mercury cycles. Such a study is now possible due to the increased availability of isotopic data (both C and Hg) from a variety of well-dated marine and terrestrial settings. Their modeling study is well-designed to address the sequence and timing of events on land vs. the oceans across the extinction boundary. I find their conclusions compelling, and am particularly encouraged by additional testable hypotheses that appear to arise from this work.

I have a few general as well as detailed comments for the authors.

- 1) A transition sentence at the end of the "Background" section (Line 88) would help bridge the discussion between the motivation for the work, and the actual description of the model which follows (Line 90). A single sentence (similar to what is stated in the abstract) would suffice here.
- 2) In the Conclusions section, the sentence that ends on Line 181 with "...and subsequent global warming" could use a reference or two, in particular with reference to the end Permian hyperthermal that coincided with the mass extinction event (maybe can cite Sun et al., 2012; or Kump, 2018, for example). A couple of references inserted here can help offset the apparently abruptly ending of this paragraph (i.e., Line 185: "At this level the marine mass extinction started"). In addition, this might be a good place to re-iterate the duration of time under discussion...is it "At this level the marine mass extinction started, approximately 60,000 years after initial disruption of terrestrial ecosystems"? (from Line 79)...or some other duration that can be defined?
- 3) Line 225 – this reference needs a date (2019 I believe)
- 4) In Figure 1, the legends for Hg (ppb) and Hg/TOC should be made larger in both panels (Marine vs. Terrestrial); they are currently too small to read
- 5) In Figure 1, need to define the NCIE here, maybe at the end of Line 317, e.g., "...and more negative deltaC13 (NCIE) values..."?
- 6) In Figure 1, should probably define EPII, as it is defined in the text (Line 76), if it is critical to the discussion... But why doesn't it appear to show up here in the deltaC13 trend? Maybe omit from this plot, if not relevant to the discussion?
- 7) In Figure 1, Line 317, is EPI synonymous with the marine extinction interval, or is it the NCIE, or are they all chronologically indistinguishable?
- 8) In Figure 3, maybe label the two columns at the top as "Volcanic" on the left, and "+Terrestrial (Biosphere)" on the right, to better orient the reader to the distinction between these 2 models?
- 9) In the Methods Summary, Line 379, what is "moly yr-1"?
- 10) In the Methods Summary, Line 390, on what basis were the ramp rate factors (0, 30, 100) chosen? Are these set to modern = 0?
- 11) In the SI, Line 410, table 1 there are several typos (should be "High latitude" not "High laittude")
- 12) In the SI, Line 429, should be "Global average surface temperature (GAST)"
- 13) In the SI, line 431, there is a typo (should be "negative" not "negativte")
- 14) In the SI, line 437, could not find Broecker and Peng (1982) in the refs
- 15) In the SI, line 514, again the flux term "moly yr-1" is not defined etc.

Once these concerns are addressed, I feel confident that readers should have a clear enough idea of how the models were constructed and be able to follow up using this approach, both for the PTME and other extinction boundaries as well. This study appears to represent a very fruitful line

of inquiry and, with the proliferation of new data from PTME sections (representing both marine and terrestrial environments), it would seem that we are starting to accelerate towards a much greater understanding of the causal mechanisms of mass extinction. This study represents a significant advance and a valuable contribution to the field, appropriate for publication in Nature Communications.

Reviewer #3 (Remarks to the Author):

Dal Corso and colleagues present evidence for a massive collapse of terrestrial ecosystems during the Permian-Triassic mass extinction (PTME). The manuscript is clear, well written, logically organized and was an enjoyable read. The authors did a nice job applying simple box models to vet a range of hypotheses. I recommend the manuscript for publication with revisions. In the revisions, I'd wish to see:

1. A more mechanistic explanation of how the authors think the terrestrial ecosystem collapse unfolded. The summary offered near the end of the paper,

"The disruption of terrestrial ecosystems started during the initial phases of the Siberian Traps emplacement due to the release of volcanic gases as CO₂, SO₂, and halogens, which could have triggered acid rain, ozone depletion, volcanic darkness, rapid cooling and subsequent global warming"

comes too late in the manuscript and is too broad. A narrower, more specific explanation buttressed by evidence would be more compelling. Is there anything in the O or S isotopic records to support the acid rain or ozone depletion hypotheses?

2. A concise, but rigorous statement about the evidence the author's have that the marine sediment isn't subject to diagenesis. This should go in the main body of the manuscript.

Specific comments:

Lines 36-38: I suggest expanding the sentence,
"The PTME is marked by an approximately 2–4-fold increase in marine sedimentary Hg concentration with respect to background levels during a ~400 kyr interval with increasingly negative $\delta^{13}\text{C}$ values, which implies..."

Lines 76-87: There have been more modern large volcanic eruptions. Is there evidence from other events that also shows C and Hg track each other in a similar to the PTME post-eruption?

Line 108: I'd express 7.6-13 Gg/yr in Mg/yr to make it easier to compare to the numbers on line 114.

Line 137: Please explain the justification for a 100-fold increase. 100 feels arbitrary.

Line 192: "...assuming that delivery of phosphate was increased by the same factor as C oxidation..." Why would it be the same? Phosphate is mostly particle bound, whereas C oxidation involves gas-phase.

RESPONSE TO REVIEWS

Reviewers' comments:

Reviewer #1 (Remarks to the Author):

Review of Dal Corso et al. "Permo-Triassic boundary carbon and mercury cycling linked to terrestrial ecosystem collapse" for Nature Communications

In this modeling study, Dal Corso et al. test several hypotheses regarding causes and timing of the end-Permian mass extinction event by focusing on the environmental effects of massive volcanic loading and terrestrial biosphere oxidation, as reflected in the global carbon and mercury cycles. Such a study is now possible due to the increased availability of isotopic data (both C and Hg) from a variety of well-dated marine and terrestrial settings. Their modeling study is well-designed to address the sequence and timing of events on land vs. the oceans across the extinction boundary. I find their conclusions compelling, and am particularly encouraged by additional testable hypotheses that appear to arise from this work.

Thanks for your positive review and useful comments.

I have a few general as well as detailed comments for the authors.

1) A transition sentence at the end of the "Background" section (Line 88) would help bridge the discussion between the motivation for the work, and the actual description of the model which follows (Line 90). A single sentence (similar to what is stated in the abstract) would suffice here.

Thanks for this comment. We added this bridge sentence: "In order to evaluate the distinct terrestrial contribution to atmosphere–ocean biogeochemistry during the PTME, separated from coeval volcanic fluxes, we present here a new biogeochemical model that couples the global Hg and C cycles."

2) In the Conclusions section, the sentence that ends on Line 181 with "...and subsequent global warming" could use a reference or two, in particular with reference to the end Permian hyperthermal that coincided with the mass extinction event (maybe can cite Sun et al., 2012; or Kump, 2018, for example).

We added the references.

A couple of references inserted here can help offset the apparently abruptly ending of this paragraph (i.e., Line 185: "At this level the marine mass extinction started"). In addition, this might be a good place to re-iterate the duration of time under discussion...is it "At this level the marine mass extinction started, approximately 60,000 years after initial disruption of terrestrial ecosystems"? (from Line 79)...or some other duration that can be defined?

Yes, the time span between the onset of the negative CIE/terrestrial disturbances and the marine mass extinction, as defined in South China (where correlation and stratigraphy of the marine-terrestrial sections is more solid), is about 60Kyr. A sentence has been added reiterating the duration of the events: "This, according to the existing chronostratigraphic framework, happened ca. 60 Kyr after the onset of the carbon-isotope perturbation and of the terrestrial ecological disturbances".

3) Line 225 – this reference needs a date (2019 I believe)

Done.

4) In Figure 1, the legends for Hg (ppb) and Hg/TOC should be made larger in both panels (Marine vs. Terrestrial); they are currently too small to read

Done.

5) In Figure 1, need to define the NCIE here, maybe at the end of Line 317, e.g., "...and more negative deltaC13 (NCIE) values...?"

We deleted NCIE, which is not necessary and was only mentioned in the text a couple of times. We change it into "carbon isotope perturbation".

6) In Figure 1, should probably define EPII, as it is defined in the text (Line 76), if it is critical to the discussion... But why doesn't it appear to show up here in the deltaC13 trend? Maybe omit from this plot, if not relevant to the discussion?

Ep. I and II indicate the two minima observed in the records as defined by Xie et al. 2007. We used this definition to allow the reader to readily observe in the figure what is explained in the text. We defined Ep. II and added text to better describe the figure caption as follows: ". After this event, the $\delta^{13}\text{C}$ record shows a second minimum (Ep. II), while Hg concentration decreases, but remains relatively higher than background values, and $\delta^{202}\text{Hg}$ and $\delta^{199}\text{Hg}$ rebound towards more positive values".

7) In Figure 1, Line 317, is EPI synonymous with the marine extinction interval, or is it the NCIE, or are they all chronologically indistinguishable?

Ep. I and II refer only to features of the $\delta^{13}\text{C}$ record. We modified the sentence introducing Ep. I and II as follows: "The $\delta^{13}\text{C}$ records at the Permian – Triassic boundary show two minima^{30,31}, which are here called EP. I and II (EP. = episode) following ref.³⁰ (Fig. 1)".

8) In Figure 3, maybe label the two columns at the top as "Volcanic" on the left, and "+Terrestrial (Biosphere)" on the right, to better orient the reader to the distinction between these 2 models?

Done.

9) In the Methods Summary, Line 379, what is "moly yr-1"?

Typo corrected to "mol yr⁻¹" meaning moles per year

10) In the Methods Summary, Line 390, on what basis were the ramp rate factors (0, 30, 100) chosen? Are these set to modern = 0?

These are multiplicative, so modern = 1 for all of them. Here 0 represents a complete collapse of terrestrial productivity, 100 is the fold change in Hg input as discussed on lines 170 and 30 is the fold change in oxidative weathering: this 30 fold increase is based on what is required to drive the C isotope excursion and on the size of the biosphere Hg flux. i.e we expect it is likely to be substantial but less than 10-fold. This is now better explained in the text by moving the methods details to be next to the associated explanation.

11) In the SI, Line 410, table 1 there are several typos (should be "High latitude" not "High laittude")

Typos corrected.

12) In the SI, Line 429, should be "Global average surface temperature (GAST)"

Corrected.

13) In the SI, line 431, there is a typo (should be “negative” not “negativte”)

Typo corrected.

14) In the SI, line 437, could not find Broecker and Peng (1982) in the refs

Added “Tracers in the Sea. WS Broecker, TH Peng. Lamont-Doherty Geological Observatory, Columbia University 2543, 125-159, 1982.”

15) In the SI, line 514, again the flux term “moly yr⁻¹” is not defined etc.

Typo corrected.

Once these concerns are addressed, I feel confident that readers should have a clear enough idea of how the models were constructed and be able to follow up using this approach, both for the PTME and other extinction boundaries as well. This study appears to represent a very fruitful line of inquiry and, with the proliferation of new data from PTME sections (representing both marine and terrestrial environments), it would seem that we are starting to accelerate towards a much greater understanding of the causal mechanisms of mass extinction. This study represents a significant advance and a valuable contribution to the field, appropriate for publication in Nature Communications.

Reviewer #3 (Remarks to the Author):

Dal Corso and colleagues present evidence for a massive collapse of terrestrial ecosystems during the Permian-Triassic mass extinction (PTME). The manuscript is clear, well written, logically organized and was an enjoyable read. The authors did a nice job applying simple box models to vet a range of hypotheses. I recommend the manuscript for publication with revisions.

Thanks for your very positive comments. We addressed all of them in the revised manuscript.

In the revisions, I'd wish to see:

1. A more mechanistic explanation of how the authors think the terrestrial ecosystem collapse unfolded. The summary offered near the end of the paper,

"The disruption of terrestrial ecosystems started during the initial phases of the Siberian Traps emplacement due to the release of volcanic gases as CO₂, SO₂, and halogens, which could have triggered acid rain, ozone depletion, volcanic darkness, rapid cooling and subsequent global warming"

comes too late in the manuscript and is too broad. A narrower, more specific explanation buttressed by evidence would be more compelling. Is there anything in the O or S isotopic records to support the acid rain or ozone depletion hypotheses?

Thanks for this comment. Some available geochemical data and palaeontological observation suggest different possible causes. For example, a single published S-isotope record from the Karoo basin in South Africa seems to suggest acid rain at the PTB, and teratological pollen/spore could indicate an increase of UV-B radiation, but the ultimate cause of the terrestrial extinction is not yet defined. We added a couple of sentences and references in the introduction to briefly illustrate some of the current hypotheses on the drivers of the terrestrial mass extinction. The new text is: “The cause of the terrestrial mass extinction is still unclear and several kill mechanisms have been

hypothesised. For example, a shift from a humid warm climate to an unstable highly seasonal climate and an associated increase in wildfires affected the equatorial Permo-Triassic peatlands, drastically reducing the abundance and diversity of the flora¹⁴; abnormal pollen and spores found in different localities around the world during the PTME interval suggest widespread mutagenesis possibly linked to an increase in UV-B radiation due to ozone depletion^{15,16}; a terrestrial S-isotope record from the Karoo basin in South Africa could indicate volcanically-driven acid rain at the P–T transition¹⁷ that might have also severely impacted the flora”.

2. A concise, but rigorous statement about the evidence the author's have that the marine sediment isn't subject to diagenesis. This should go in the main body of the manuscript.

Thanks for this comment. Indeed, in some P-T sections, as expected in deep time sequences, carbonates can show different degrees of alteration and so their original $\delta^{13}\text{C}$ signature may not be fully pristine. This is for example true for a record at Meishan that shows very negative “anomalous” values, which are not in the record by Xie et al. 2007 we used. The record by Xie et al. in our Fig. 1 has been replicated in many different sections and is seen in $\delta^{13}\text{C}$ data from different substrates (carbonates, bulk organic matter and plant material). This fact indicates the C-isotope changes record isotopic changes in the reservoirs of the global C-cycle. We added a sentence and references to strengthen this point: “The observed $\delta^{13}\text{C}$ trends are similarly recorded in different depositional settings and by different substrates (carbonates, bulk organic matter, separate plant remains)^{14,30–32}, strongly indicating that they represent actual changes in the C-isotope composition of the reservoirs of the exogenic C-cycle”.

Specific comments:

Lines 36-38: I suggest expanding the sentence,

“The PTME is marked by an approximately 2–4-fold increase in marine sedimentary Hg concentration with respect to background levels during a ~400 kyr interval with increasingly negative $\delta^{13}\text{C}$ values, which implies...”

We expanded the sentence as follows: “...values, which implies a relatively long-term injection of Hg and ^{13}C -depleted CO_2 into the atmosphere – land – ocean system during this time”.

Lines 76-87: There have been more modern large volcanic eruptions. Is there evidence from other events that also shows C and Hg track each other in a similar to the PTME post-eruption?

Thanks for this comment. The records are not as good as for the PTME. We added the following sentence on the conclusions to make this point: “For other events the Hg records are not so consistent nor as detailed as for the PTME. However, it is very likely that future research on other intervals could show the same Hg and C patterns as for the PTME”.

Line 108: I'd express 7.6-13 Gg/yr in Mg/yr to make it easier to compare to the numbers on line 114.

Done.

Line 137: Please explain the justification for a 100-fold increase. 100 feels arbitrary.

This is our estimation of the Hg flux required to drive the magnitude of change observed, and is consistent with available biosphere and soil Hg. Our choice of a round number reflects the uncertainty here. We have made this clearer in the text: “This level of increase represents the magnitude required to drive the sedimentary Hg signal that we observe, and is compatible with the available terrestrial biosphere Hg reservoir”.

Line 192: "...assuming that delivery of phosphate was increased by the same factor as C oxidation..."
Why would it be the same? Phosphate is mostly particle bound, whereas C oxidation involves gas-phase.

Agreed, it is likely to be less. We have made this clearer by simply referencing the changes to P input that are expected to drive anoxia (~2-3 fold): "..., but other models have shown that a relatively small increase in marine P delivery (2–3 fold) has the potential to drive marine anoxia or euxinia^{52,53}".

REVIEWERS' COMMENTS:

Reviewer #1 (Remarks to the Author):

The authors have addressed all of my concerns. The revised version of the manuscript reflects a high level of attention to detail, and the authors are to be commended for an excellent job overall. I can now wholeheartedly recommend this study for publication.

Reviewer #3 (Remarks to the Author):

The authors have done a great job responding to reviewer comments. This paper is ready to publish. I look forward to seeing it in the literature.

REPLY TO REVIEWERS' COMMENTS:

Reviewer #1 (Remarks to the Author):

The authors have addressed all of my concerns. The revised version of the manuscript reflects a high level of attention to detail, and the authors are to be commended for an excellent job overall. I can now wholeheartedly recommend this study for publication.

Reviewer #3 (Remarks to the Author):

The authors have done a great job responding to reviewer comments. This paper is ready to publish. I look forward to seeing it in the literature.

We would like to thank very much the reviewers for their very useful comments that greatly improved the manuscript!